# Study on Hot Tensile Deformation Behavior and Hot Stamping Process of GH3625 Superalloy Sheet

**DOI:** 10.3390/ma16051927

**Published:** 2023-02-25

**Authors:** Shixin Peng, Jie Zhou, Jie Peng, Heping Deng, Fanjiao Gongye, Jiansheng Zhang

**Affiliations:** 1School of Materials Science and Engineering, Chongqing University, Chongqing 400044, China; 2Chongqing Jiepin Technol Co., Ltd., Chongqing 400044, China

**Keywords:** GH3625 superalloy, grain growth, work hardening model, modified Arrhenius model, hot stamping

## Abstract

Hot tensile tests of the GH3625 superalloy were carried out under the temperature range of 800–1050 °C and strain rates of 0.001, 0.01, 0.1, 1, and 10 s^−1^ on a Gleeble-3500 metallurgical processes simulator. The effect of temperature and holding time on grain growth was investigated to determine the proper heating schedule of the GH3625 sheet in hot stamping. The flow behavior of the GH3625 superalloy sheet was analyzed in detail. The work hardening model (WHM) and the modified Arrhenius model, considering the deviation degree R (R-MAM), were constructed to predict the stress of flow curves. The results showed that WHM and R-MAM have good prediction accuracy by evaluating the correlation coefficient (R) and the average absolute relative error (AARE). Additionally, the plasticity of the GH3625 sheet at elevated temperature drops with the increasing temperature and decreasing strain rate. The best deformation condition of the GH3625 sheet in the hot stamping is in the range of 800~850 °C and 0.1~10 s^−1^. Finally, a hot stamped part of the GH3625 superalloy was produced successfully, which had higher tensile strength and yield strength than the as-received sheet.

## 1. Introduction

Owing to the increasing demand of the thrust-to-weight ratio of aero-engines, the material of the flame tube was changed from the stainless steel AISI310 to the Ni-based superalloy with solid solution strengthening or precipitation strengthening [1,2,3,4]. The GH3625 superalloy sheet can be used for manufacturing the flame tube, which has excellent high-temperature strength and oxidation resistance, outstanding creep and thermal fatigue strength [5,6,7,8]. The GH3625 is a Ni-Cr-Mo-Nb multicomponent alloy, which is a Chinese brand and similar to Inconel 625. The GH3625 superalloy belongs to wrought superalloys with solid solution strengthening. 

Hot stamping technology can be adopted to form a GH3625 superalloy sheet due to high strength and microhardness. The characteristics of hot working in superalloys are low plasticity, high deformation resistance, narrow processing temperature range, no phase recrystallization and low thermal conductivity [9,10,11,12,13]. The current research on GH3625 or Inconel 625 mainly focuses on the hot deformation and dynamic recrystallization behavior of forged or casted bars during hot compression [14,15]. Jia et al. [16] studied the hot deformation behavior and the microstructural characteristics of Inconel 625, and the optimum condition of the as-cast Inconel 625 alloy was determined at a temperature range of 1100–1200 °C and strain rate of 1–10 s^−1^, where the perfect dynamic recrystallization occurs and fine grain structure is obtained. Maj et al. [17] investigated and analyzed Portevin Le-Chatelier effect (PLC effect) of the Inconel 625 superalloy using tensile tests and compression tests, and the results showed that cross-core diffusion was most probably responsible for the PLC effect in Inconel 625. Li et al. [18] investigated the microstructure evolution and nucleation mechanisms of dynamic recrystallization for hot-deformed Inconel 625 superalloy and found that the dominant nucleation mechanism of dynamic recrystallization at 1150 °C is the discontinuous dynamic recrystallization. Chen et al. [19] studied hot deformation behavior and the microstructural evolution of Inconel 625 superalloy plates in a range of working temperatures (800–1050 °C) and strain rates (0.001–1 s^−1^). Gao et al. [20] investigated the effect of twin boundaries on the microstructure and mechanical properties of the Inconel 625 alloy during hot deformation, the result indicated that the twin boundaries can improve strength and ductility. Liu et al. [21] studied the temperature effect on the deformation behavior, microstructural evolution and fracture mechanism of the Inconel 625 sheet from room temperature to 950 °C. Badrish et al. [22] studied the anisotropic behavior and mechanical properties of the Inconel 625 superalloy sheet from room temperature to 600 °C. Godasu et al. [23] investigated the effects of strain rate on thermal deformation behavior and microstructure evolution of Inconel 625 superalloy at 1000 °C through plane strain compression (PSC) tests. However, there is minimal literature about the hot stamping process and the tensile behavior of the GH3625 or Inconel 625 superalloy sheets at higher temperatures and stain rates. Therefore, a series of experiments and investigations must be conducted in order to confirm the forming process and manufacture the GH3625 product.

In this paper, the grain growth behaviors of the GH3625 sheet during heating were investigated to determine the proper heating schedule before forming. The tensile test of the GH3625 sheet at different temperatures and strain rates were conducted on a Gleeble-3500 metallurgical processes simulator. A constitutive model of different flow curves of the GH3625 sheet was constructed. The high temperature plasticity of the GH3625 sheet was studied. Finally, a hot stamped part of the GH3625 superalloy was produced successfully, mechanical properties and the microstructure of the stamped piece were tested and observed. The paper will be helpful in providing a theoretical basis and engineering guidance for the hot stamping process of nickel-based superalloy sheets.

## 2. Experimental Procedures

The GH3625 superalloy sheet used in this study had a nominal thickness of 2 mm. The chemical composition is listed in Table 1. The microstructure of the as-received GH3625 sheet under the solution-heat-treated condition is shown in Figure 1. Figure 1a shows the Euler angle diagram, which is composed of equiaxed grains and annealing twins. The color distribution of each grain is random, which indicates a weak grain orientation. The average grain size of the alloy is around 36μm. The orientation imaging microscopy (OIM) map is shown in Figure 1b, in which Σ3, Σ9 and Σ27 boundaries are represented by red lines, fuchsia lines and green lines, respectively. A misorientation angle greater than 15° is defined as high angle grain boundaries (HAGBs), which is represented by red lines. The misorientation angles in the range of 2~15° are defined as low angel grain boundaries (LAGBs), which are represented as green lines. The fractions of the Σ3, Σ9 and Σ27 boundaries are 42%, 1% and 0.4%, respectively. Figure 1c shows the misorientation angel distribution. It can be seen that a large number of orientation angles are 60°, with the rest mainly distributed between 10° and 60°. Figure 1d shows the angle distribution of orientation difference from point A to B. Six red lines of grain boundaries were present from point A to B, and the orientation difference angle of each twin was 60°, indicating that the twins have a <111>/60° orientation difference relationship with the parent grain.

The samples with sizes of 10 mm × 15 mm × 2 mm were obtained by wire electrode cutting. These specimens were then heat-treated in the temperature range of 950–1250 °C with an interval of 50 °C, and then held, respectively, for 5, 15, 30, 45 and 60 min in the atmosphere box furnace [3]. Subsequently, these specimens were water-quenched to preserve the high-temperature microstructure. The hot tensile tests were conducted on a Gleeble-3500 metallurgical processes simulator at temperatures of 800, 850, 900, 950, 1000 and 1050 °C and strain rates of 0.001, 0.01, 0.1, 1 and 10 s^−1^. All tensile specimens were heated to predetermined temperatures at 10 °C/s and held for 3 min before stretching deformation. The hot tensile specimens which conform to the American standard E21-20 were machined into a certain shape shown in Figure 2. The microstructures were examined by Axiovert 200 MAT OM and JSM-7000F SEM. The corrodent, comprising 11 mL ethyl alcohol + 14 mL hydrochloric acid + 1.4 g copper sulfate, was applied to the metallographic acquisition, with a corrosion time of 9 min. The samples for an electron back-scattered diffraction (EBSD) investigation were electropolished with a solution of 10% perchloric acid in ethanol at 20 V for 30 s below −30 °C. EBSD maps were obtained with a step size of 1 μm.

## 3. Results and Discussion 

### 3.1. Effect of Temperature and Holding Time on Grain Growth

The heating temperature and holding time are critical parameters for the hot stamping process of the GH3625 sheet. The microstructural evolution of the GH3625 superalloy at 1050 °C for various holding times is shown in Figure 3; it can be seen that grain is fine and uniform, and grain growth is not obviously connected with holding time. Figure 4 displays the microstructures of the GH3625 superalloy at 5 min for various temperatures, and it can be seen that the grain grew rapidly and the mixed grain structure with large size differences appeared when the temperature reached 1150 °C. This is because the secondary phase particles on grain boundary was dissolved and lost the pinning effect above 1100 °C, which greatly promoted grain growth [3]. Subsequently, the grain size was further coarsened and became more uniform when the temperature was raised to 1250 °C. 

In order to further uncover the influence of temperature and holding time on average grain size, Figure 5 shows the variation diagram of grain size with holding time under different temperatures. In Figure 5a, the grain size increased rapidly in the first five minutes, and then grain growth became sluggish with holding time, which is due to the fact that grain growth is characterized by grain boundary migration, which is a time-consuming process. There were large number of fine grains at the beginning of holding time, which provide a strong driving force for the grain growth. However, the further migration of grain boundaries becomes difficult because of the reduction of the grain boundaries energy [24]. In addition, Figure 5b shows that the color map changes suddenly from blue to red when the temperature exceeds 1100 °C. However, the color map changes gently with the extension of holding time. Therefore, heating temperature plays a more dominant role in influencing grain size than holding time.

The grain growth rate and grain expansion rate were defined in order to quantitatively describe the effects of heating time and heating temperature on grain size. Grain growth rate was defined as the change of average grain size per unit time under a certain heating temperature, and its unit is μm/s. The grain expansion rate was defined as the average grain size change caused by the change of unit temperature under a certain holding time, and its unit is 1/°C. Figure 6a shows the curve of grain size growth rate at different temperatures, the grain growth rate reaches the maximum value (0.75 μm/s) at the range of 0~300 s and then decreases sharply. As shown in Figure 6b, the grain expansion rate is suddenly increased in the range of 1150~1200 °C. In conclusion, the heating temperature of the GH3625 sheet should not be higher than 1100 °C in order to avoid mixed structure and coarse grains in hot stamping. The heating temperature and holding time of the GH3625 sheet can be set at a range of 950–1100 °C and 5 min, respectively, in order to obtain fine grain size and high production efficiency.

### 3.2. Grain Growth Model of GH3625 during Heating

The classical Arrhenius model can be used to describe the grain growth behavior when the heating temperature is 1100~1250 °C and the holding time is 5~45 min. The expression of the classical Arrhenius model is as follows [3]: (1)$$d-d_{0}=Bt^{m}exp[-Q_{d}/RT]$$
where d is the grain size (μm), $d_{0}$ is initial grain size (36μm), t is holding time (s), T heating temperature (K) and R is the gas constant (8.314 J/(mol·K)). $Q_{d}$ is the activation energy of grain growth (J/mol), m is time exponent and B is the material constant. It can be obtained by taking the natural logarithm of Equation (1):(2)$$\ln(d-d_{0})=\ln B+m\ln t-Q_{d}/\left(RT\right)$$

For a constant temperature T in Equation (2), the partial derivative of grain size with respect to time can be obtained:(3)$$m=\partial \ln(d-d_{0})/\partial \ln t$$

For a constant holding time t in Equation (2), the partial derivative of grain size with respect to temperature can be obtained:(4)$$Q_{d}=-R\partial \ln(d-d_{0})/\partial \ln(1/T)$$

According to the calculation results of Equations (2)–(4), Figure 7 shows the linear fitting relationship between different parameter combinations, the values of m, B and $Q_{d}$ are obtained. At last, the grain growth model of the GH3625 alloy is as follows: (5)$$d=d_{0}+9.86\times 10^{10}t^{0.09}exp[-259272/RT]$$

In order to evaluate the prediction accuracy of Equation (5), Figure 8 shows the comparison of prediction results of grain size with experimental results. The average absolute relative error (AARE) is 5.8174%, which indicates that the model has good prediction precision.

### 3.3. Hot Tensile Deformation Behavior

The true stress–strain curves of the GH3625 superalloy at different strain rates are shown in Figure 9. It is a universal phenomenon that true stress decreases with increasing temperature and decreasing strain rate. As shown in Figure 9, all of these flow curves can be classified into four types: work hardening (WH), transitional dynamic recovery (TDRV), dynamic recovery (DRV) and dynamic recrystallization (DRX). The flow stress continues to rise with the increasing strain until the fracture occurs rapidly, which is termed as WH. The flow stress tends to be basically stable at a peak strain when a balance between dynamic softening and work hardening is reached, which is known as DRV. The flow stress drops rapidly after peak strain and then remains unchanged, which is known as DRX. TDRV is defined as the instable transitional status from WH to DRV or DRX. 

Every flow curve in Figure 9 is indicated by arrows as the corresponding type of curve (WH, TDRV, DRV and DRX). Serration flow during the hot tensile deformation of the GH3625 was observed at all flow curves except DRX curves, signifying the occurrence of dynamic strain ageing (DSA). DSA is caused by the interaction between solute atoms and mobile dislocations during straining [25]. The amplitude of variations for serration types A and B is around 10 MPa and 20 MPa, respectively. Moreover, a sharp drop in flow stress is also observed (indicated by circles 1~5), the decrease range is 18 MPa, 27 MPa, 11 MPa, 40 MPa and 100 MPa from circle 1 to circle 5, respectively. It is noteworthy that TDRV curves were characterized in the form of type A or type B or by a sharp drop in flow stress.

It is obvious that WH occurs at 800~1000 °C and 1~10 s^−1^ or 800~900 °C and 0.1~1 s^−1^. The hardening flow stress of WH has a linear relationship with strain. The strain hardening exponent remains basically constant with temperature and is a functional relationship with strain rate. Additionally, the yield strength remains basically stable in conjunction with the strain rate for WH curves, and the flow behavior is independent of strain rate at 800~950 °C and 1~10 s^−1^. Therefore, the hardening curves of the WH model (WHM) can be expressed as follows:(6)$$\sigma =n\varepsilon +b_{0}$$
(7)$$n=f_{n}\left(\dot{\varepsilon }\right),\;b_{0}=f_{0}\left(T\right)$$
where σ is the flow stress (Mpa), $\dot{\varepsilon }$ is the strain rate (s^−1^), n is the hardening exponent (Mpa), ε is strain, b_0_ is the intercept of hardening curves with removing elastic strain (Mpa) and T is the temperature (°C). $\;\;f_{n}\left(\dot{\varepsilon }\right)$ is a function of strain rate and hardening exponent, and $\;f_{0}\left(T\right)$ is a function of the temperature and the intercept of hardening curves. 

The fitted curves of n and $b_{0}$ are shown in Figure 10, and the coefficients of the third order polynomial functions are also listed in Figure 10. The adjusted coefficients of determination between the fitted results and n and $b_{0}$ are 0.9616 and 0.9635, respectively, which revealed the excellent degree of fitting. Thus, the flow stress of the WH type can be calculated. Figure 11 shows the prediction results of the hardening curves of WH types. In Figure 11, the curves of different colors is experimental results at different temperatures, the hollow, square points of different colors are the predicted results of the corresponding color curve. The correlation coefficient (R) and AARE were used to evaluate the model of WH. Figure 12 shows the comparison between predicted data and experimental data. The R-values and AARE values of the WH model are 0.9904 and 2.5041%, respectively, which indicates that WHM has good prediction precision.

The other flow curves, which include TDRV, DRV and DRX, can be described using the Arrhenius model, and its relevant mathematical expressions are as follows:(8)$$\dot{\varepsilon }=A{\left[\sinh\left(\alpha \sigma \right)\right]}^{n}\exp\left[-Q/\left(RT\right)\right]$$
(9)$$Z=\dot{\varepsilon }\exp\left[-Q/\left(RT\right)\right]=A{\left[\sinh\left(\alpha \sigma \right)\right]}^{n}$$
(10)$$\sigma =\frac{1}{\alpha }\ln\{(\frac{Z}{A}{)}^{\frac{1}{n}}+[(\frac{Z}{A}{)}^{\frac{2}{n}}+1{]}^{\frac{1}{2}}\}$$
where T is the temperature (K), Q is the deformation activation energy (KJ/mol), R is gas constant (8.314 J·mol^−1^·K^−1^), Z is strain rate factor of temperature compensation, and the remaining values (α,A,n,Q) are material constants, which are related to the strain. Equation (10) is the deformation expression of Equation (8), which can directly calculate the flow stress. 

The nine flow curves at temperatures of 950 °C,1000 °C and 1050 °C and strain rates of 0.001, 0.01 and 0.1 s^−1^ were chosen as the experimental data for the establishment of the Arrhenius model. The constants (α, lnA,n and Q) can be obtained by calculating the average slopes of fitted lines and the expression Q=1000Rnk [26,27]. The four material constants at a strain ranging from 0.015–0.20 with an interval of 0.05 were calculated and then fitted by a 8th order polynomial. Figure 13 shows the fitted curves of the four material constants and the coefficients of the eighth order polynomial functions. The adjusted coefficients of determination (Adj-R^2^) between fitted curves and α, n,lnA,Q are 0.9967, 0.9984, 0.9793 and 0.9791, respectively, which revealed a good fitting degree of fitted curves. So far, the flow stress of all TDRV, DRV and DRX curves under a given strain, temperature and strain rate can be predicted.

According to the calculation results of Arrhenius model, there are deviations in varying degree between the experimental results and the prediction results. Figure 14 only shows the deviations at $\dot{\varepsilon }$ = 0.001 and $\dot{\varepsilon }$ = 0.01. In Figure 14, the deviations in the y direction are large and decrease with the increasing temperature, but the variation trend of the prediction points is consistent with the experiment curves. The deviation in the y direction can be defined by the following equation:(11)$$R=\sigma_{e}/\sigma_{p}$$
where R is the deviation degree, $\sigma_{e}$ is the maximum value of experiment curve and $\sigma_{p}$ is the maximum value of prediction point. It is obvious that the smaller the R value, the greater the degree of deviation and vice versa.

The R-values of all TDRV, DRV and DRX curves are shown in Table 2. The R-value rises with increasing temperatures and decreasing strain rate. Moreover, the R-value basically remained unchanged along with the strain rate at a strain rate of 0.001–0.1 s^−1^, which implies that the R-value is only dependent on the temperature in this range. However, the R-value displayed great changes at a strain rate range of 1–10 s^−1^, which demonstrates that the R-value is dependent on both the temperature and temperature in the range. 

Consequently, a modified Arrhenius model, which takes into account the deviation degree R (which is termed as R-MAM), can be denoted as follows: (12)$$\sigma =R\frac{1}{\alpha }\ln\{(\frac{Z}{A}{)}^{\frac{1}{n}}+[(\frac{Z}{A}{)}^{\frac{2}{n}}+1{]}^{\frac{1}{2}}\}$$
(13)$$R=R_{1}R_{2}=f_{1}\left(T\right)f_{2}\left(\dot{\varepsilon }\right),R_{1}=f_{1}\left(T\right),\;R_{2}=f_{2}\left(\dot{\varepsilon }\right)$$

In which R_1_ and R_2_ are the deviation degree with respect to temperature and strain rate, respectively. The fitted curves and polynomial coefficients of R_1_ and R_2_ are shown in Figure 15. The Adj-R^2^ between fitted curves and R_1_ and R_2_ are 0.9947 and 0.9999, respectively. Figure 16 shows the prediction results of the TDRV, DRV and DRX curves calculated by R-MAM, and the comparison between predicted data and experimental data is shown in Figure 17. The values of R and AARE are 0.9695 and 5.530%, respectively. This also indicates that R-MAM has the better prediction precision for DRX than for DRV or TDRV.

### 3.4. Plasticity at High Temperatures

The elongations at break for all samples and fractured tensile samples are presented in Figure 18. It can be seen that the elongation at break varies from 49% to 56% and the GH3625 sheet has good formability when the deformation conditions are in the range of 800~850 °C and 0.1~10 s^−1^. Moreover, the plasticity drops with the increasing temperature and decreasing strain rate, which is the opposite to what occurs when using general metal materials. Kong et al. [28] also obtained similar results. As shown in Figure 19, the decrease in the ductility is mainly attributed to the increased precipitation of second-phase particles at higher temperatures and lower strain rates. These second-phase particles at the grain boundaries originated from the supersaturated solid–solution substrate, which reduced plasticity and increased strength.

### 3.5. Production Verification 

The whole hot stamping process of the GH3625 sheet was divided into three main process, namely the blank transfer stage, the forming stage and air cooling [29,30,31]. The dimensions of the rectangular billet with a central hole of φ80 mm was 500 mm × 330 mm × 2 mm, and the billet was heated to 1000 °C and quickly transferred to the mold. Finally, a hot stamped part of the GH3625 superalloy was successfully produced by adopting the hot stamping process parameters listed in Table 3. Figure 20 shows the mechanical properties and microstructure of the GH3625 stamping part. In Figure 20a, a miniature dog bone specimens were machined due to the geometrical constraints. The miniature dog bone specimen matched the response of ASTM specimens at both low and high strain rates [32], and the 2D graph of dog bone specimen is shown in Figure 20c. In Figure 20b, the stress–strain curves of samples 1# and 2# are similar. The average yield strength, tensile strength and elongation at break of the two samples are 544 MPa, 1556 MPa and 60%, respectively. In comparison with the as-received sheet, the yield strength and tensile strength of the stamped parts are increased by 65% and 17%, respectively, and the elongation at break is decreased by 6%. This is mainly due to the precipitation of second phase particles in the workpiece during the hot stamping process, as shown in Figure 20d. In addition, Figure 20e shows that the microstructure of the stamped part is uniform and fine-grained, and the average grain size is about 45μm. 

## 4. Conclusions

The hot tensile tests of the GH3625 superalloy under the temperature range of 800–1050 °C with an interval of 50 °C and strain rates of 0.001, 0.01, 0.1, 1 and 10 s^−1^ were conducted on a Gleeble-3500 metallurgical processes simulator and the true stress–strain curves were obtained. The effect of holding time and temperature on grain growth was investigated, and the model of grain growth was constructed. The constitutive model of different flow curves for the GH3625 sheet was developed. A hot stamped part of the GH3625 superalloy was produced successfully. The following main conclusions can be drawn from this work:The grain size of the GH3625 superalloy increased rapidly, and a mixed structure appeared when the temperature reached 1150 °C. The heating temperature played a greater influence on grain growth than the holding time. During the hot stamping process, the heating temperature and holding time of the GH3625 sheet can be set to the range of 950–1100 °C and 5 min, respectively.The work hardening model (WHM) was developed to predict WH curves. A modified Arrhenius model, taking the deviation degree R (R-MAM) into account, were developed in order to predict the DRX, DRV or TDRV curves. The results demonstrated that both models have good prediction accuracy for separate flow curves.The GH3625 sheet has good formability when the deformation conditions are in the range of 800~850 °C and 0.1~10 s^−1^. Moreover, the plasticity of the GH3625 sheet drops with the increasing temperature and decreasing strain rate, which is mainly attributed to the increased precipitation of second-phase particles at higher temperatures and lower strain rates. In comparison with the as-received sheet, the yield strength and tensile strength of the stamping parts increased by 65% and 17%, respectively, and the elongation at break was decreased by 6%.

## Figures and Tables

**Figure 1 materials-16-01927-f001:**
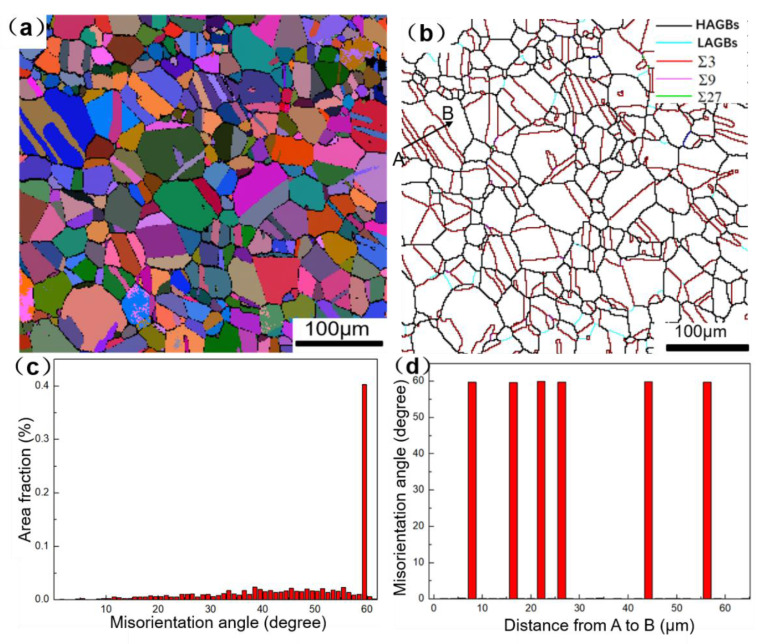
The microstructure of the GH3625 with as-received status: (**a**) all Euler diagrams; (**b**) OIM map; (**c**) misorientation angle distribution map; (**d**) misorientation angle from A to B.

**Figure 2 materials-16-01927-f002:**
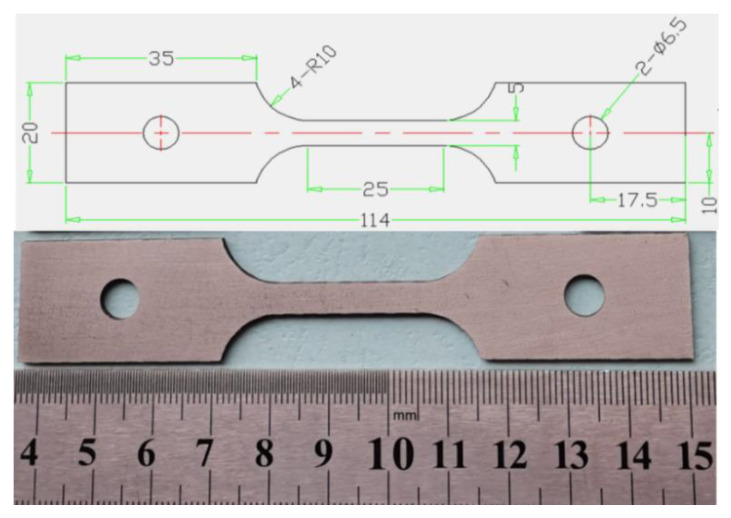
The tensile specimen of the GH3625 sheet.

**Figure 3 materials-16-01927-f003:**
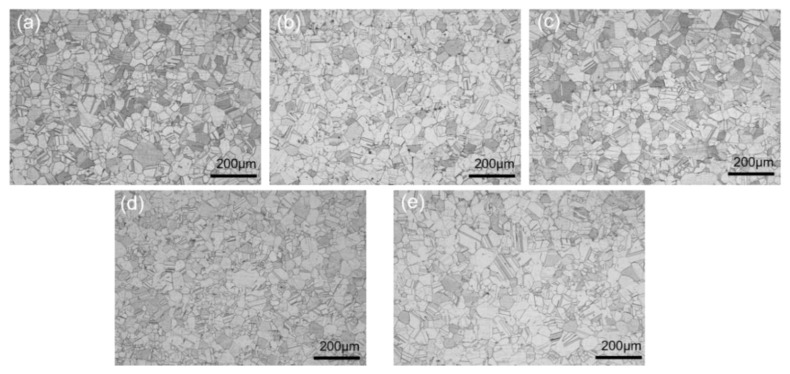
OM of the GH3625 superalloy at 1050 °C for various times: (**a**) 5 min; (**b**) 15 min; (**c**) 30 min; (**d**) 45 min; (**e**) 60 min.

**Figure 4 materials-16-01927-f004:**
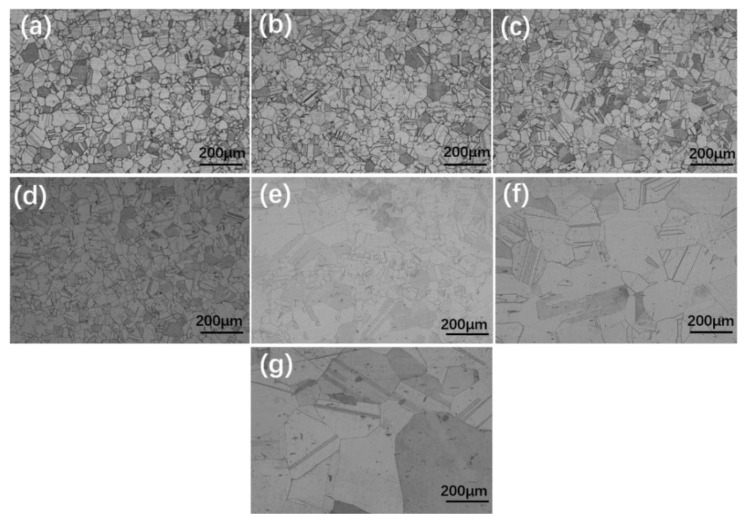
OM of the GH3625 superalloy at different temperatures for 5 min: (**a**) 950 °C; (**b**) 1000 °C; (**c**) 1050 °C; (**d**) 1100 °C; (**e**) 1150 °C; (**f**) 1200 °C; (**g**) 1250 °C.

**Figure 5 materials-16-01927-f005:**
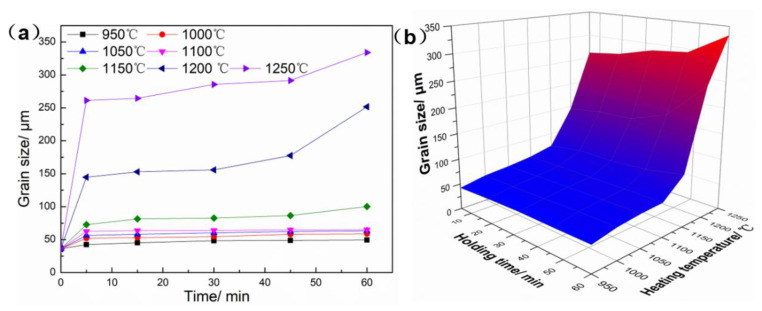
Variation diagram of grain size for the GH3625 at different heating temperatures and holding times: (**a**) 2D curves, (**b**) 3D color map.

**Figure 6 materials-16-01927-f006:**
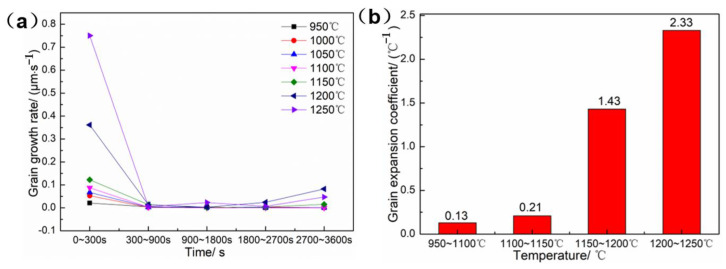
The grain growth rate (**a**) and the grain expansion coefficient (**b**) of the GH3625 sheet.

**Figure 7 materials-16-01927-f007:**
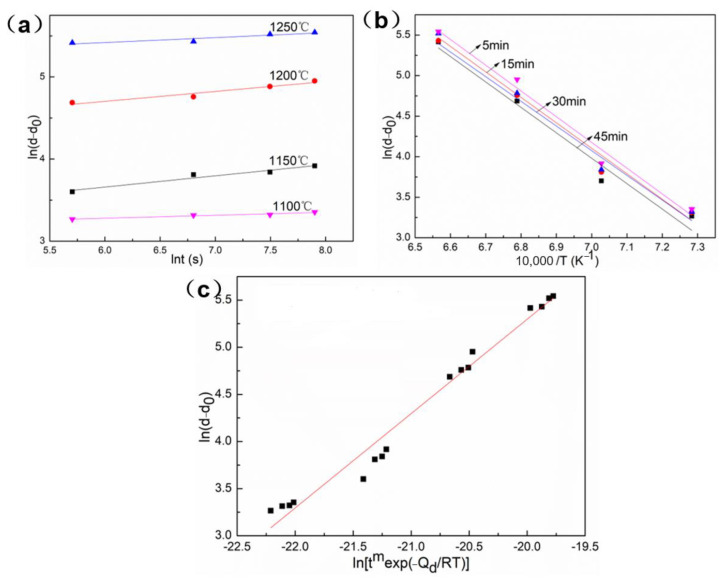
The linear fitting relationship of different parameter combinations: (**a**) lnt−ln(d−d_0_), (**b**)1000/T−ln(d−d_0_), (**c**) ln[t^m^exp(−Q_d_/RT)] − ln(d − d_0_).

**Figure 8 materials-16-01927-f008:**
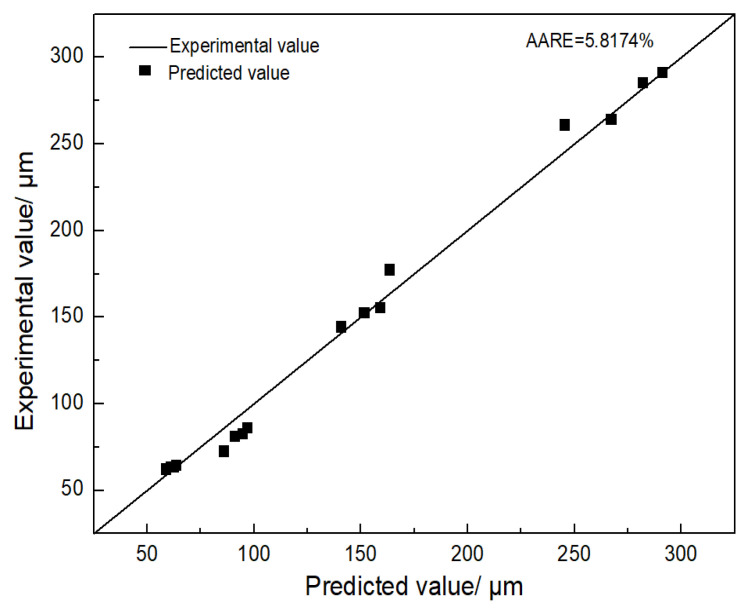
The comparison of the prediction results of grain size with experimental results.

**Figure 9 materials-16-01927-f009:**
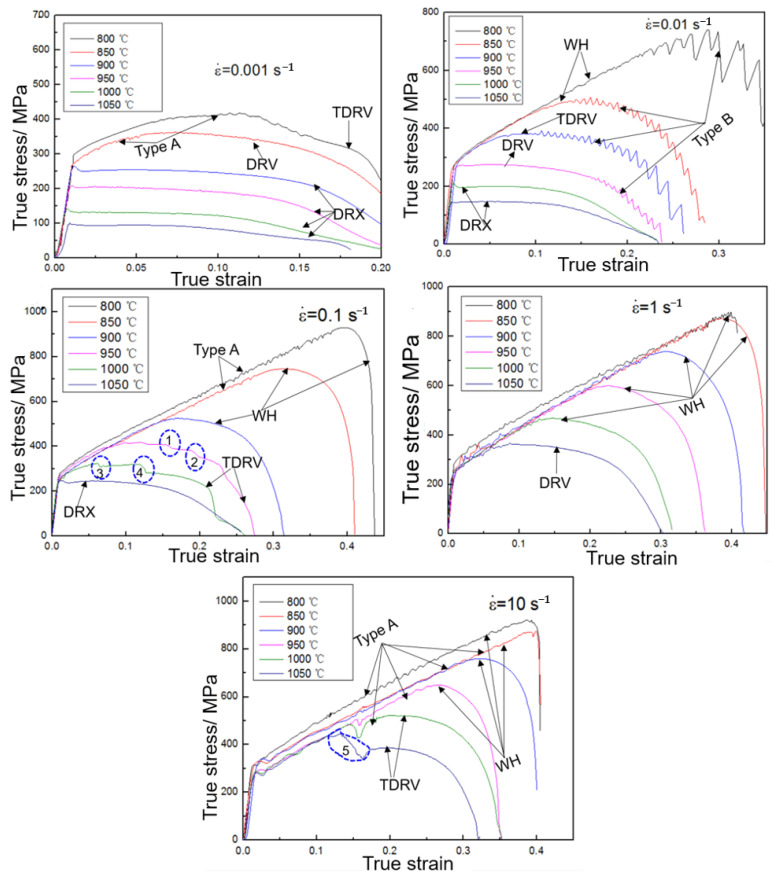
The true stress strain curves of the GH3625 superalloy at different strain rates.

**Figure 10 materials-16-01927-f010:**
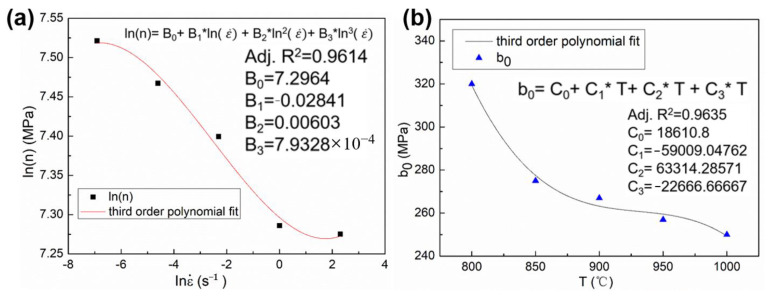
The fitted curves of the (**a**) ln(n) and (**b**) b_0_.

**Figure 11 materials-16-01927-f011:**
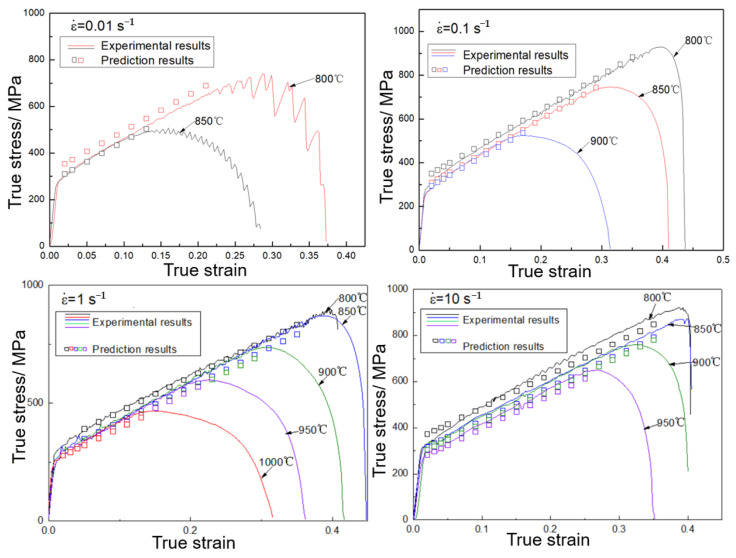
The prediction results of the hardening curves of WH types.

**Figure 12 materials-16-01927-f012:**
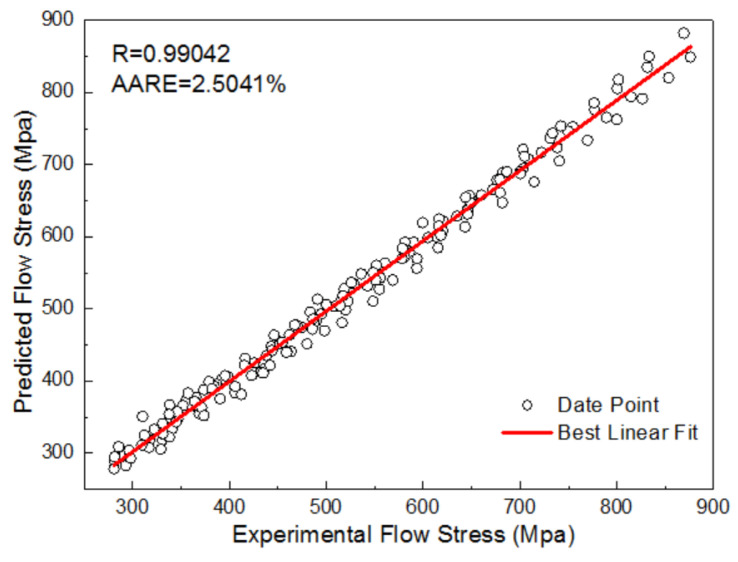
The comparison between the predicted data and the experimental data by WHM.

**Figure 13 materials-16-01927-f013:**
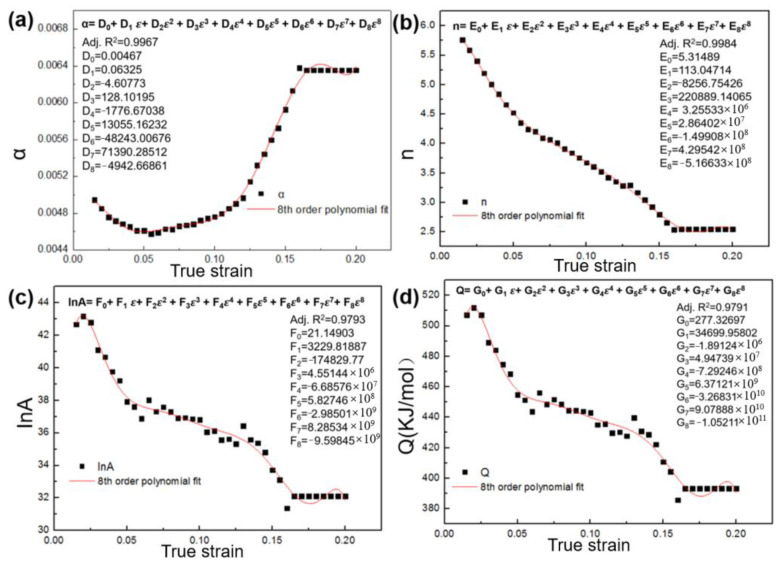
The fitted curves of the materials constant: (**a**) α; (**b**) lnA; (**c**) n; (**d**) Q.

**Figure 14 materials-16-01927-f014:**
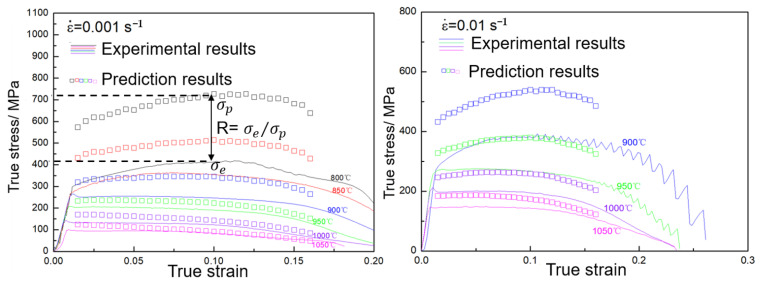
The prediction results of the DRX, DRV and TRV types by the Arrhenius model.

**Figure 15 materials-16-01927-f015:**
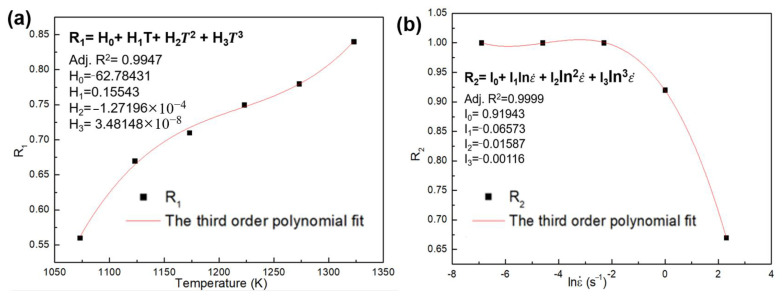
The fitted curves of the (**a**) R_1_ and (**b**) R_2_.

**Figure 16 materials-16-01927-f016:**
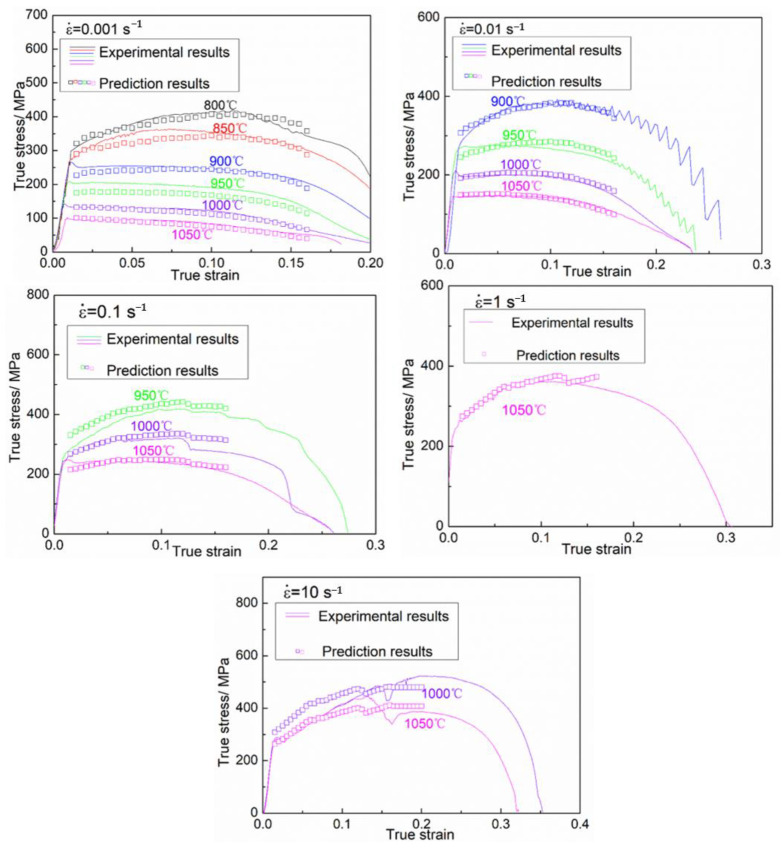
The prediction results of DRX, DRV and TDRV types by R-MAM.

**Figure 17 materials-16-01927-f017:**
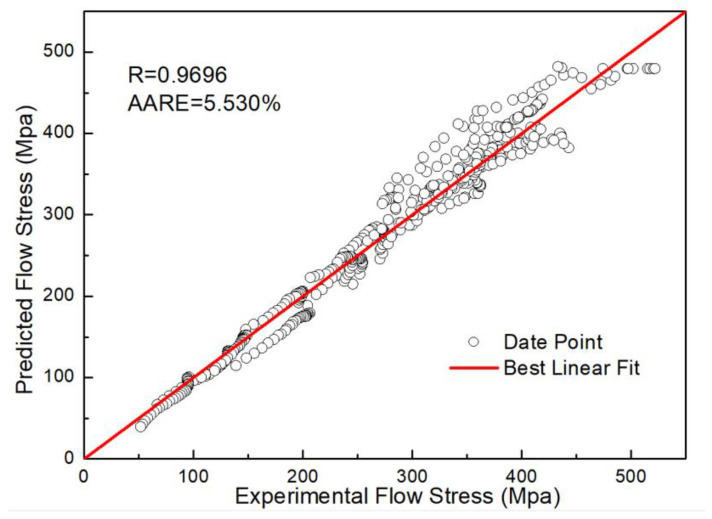
The comparison between predicted data and experimental data by R-MAM.

**Figure 18 materials-16-01927-f018:**
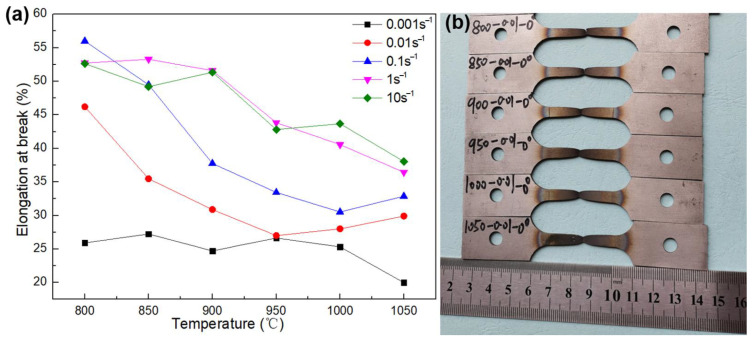
The elongation at break of all samples (**a**) and the fractured tensile samples (**b**).

**Figure 19 materials-16-01927-f019:**
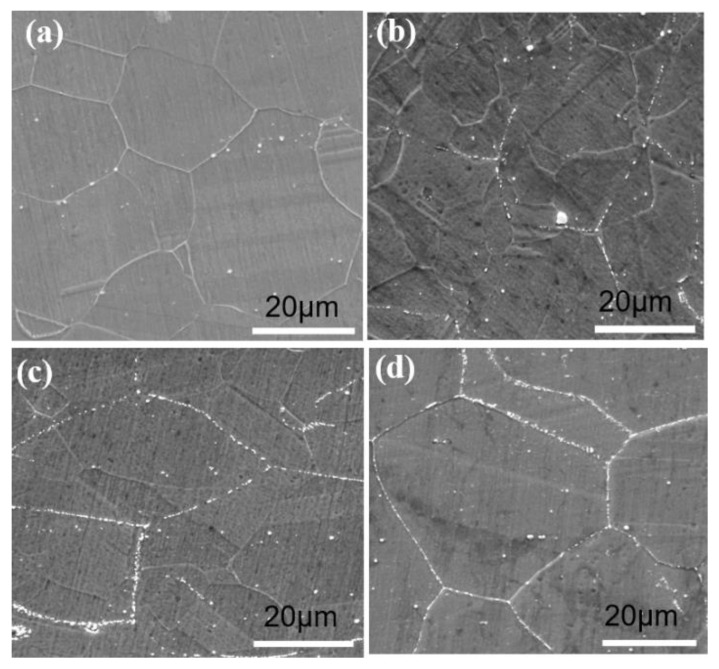
The precipitation of second-phase particles at different temperatures: (**a**) as-received status; (**b**) 800 °C; (**c**) 900 °C; (**d**) 1000 °C.

**Figure 20 materials-16-01927-f020:**
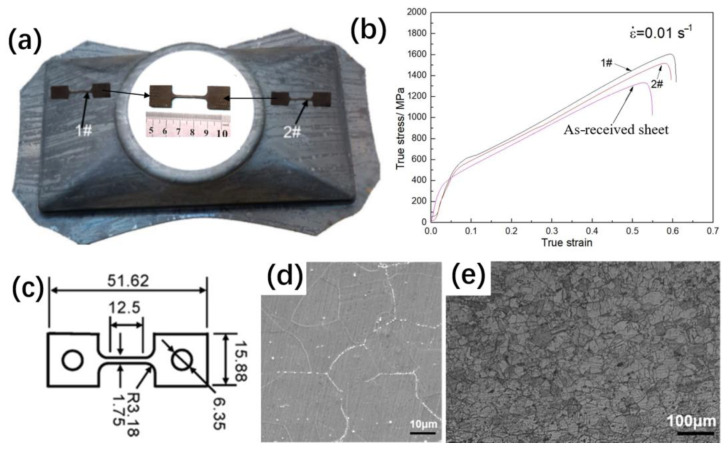
Mechanical properties and microstructure of the GH3625 stamping part: (**a**) stamped part and sampling locations; (**b**) true stress–strain curves; (**c**) 2D graph of tensile specimen; (**d**) SEM; (**e**) OM.

**Table 1 materials-16-01927-t001:** Chemical composition of the GH3625 superalloy (mass%).

Cr	Mo	Nb	Fe	C	Al	Ti	Mn	S	Ni
21.5	9	3.6	2	0.05	0.2	0.2	0.2	0.001	Bal

**Table 2 materials-16-01927-t002:** The R-value at different temperatures and strain rates.

	0.001 s^−1^	0.01 s^−1^	0.1 s^−1^	1 s^−1^	10 s^−1^
1073 K	0.56	--	--	--	--
1123 K	0.69	0.66	--	--	--
1173 K	0.71	0.71	--	--	--
1223 K	0.83	0.71	0.71	--	--
1273 K	0.84	0.76	0.75	--	0.49
1323 K	0.90	0.81	0.81	0.75	0.59

**Table 3 materials-16-01927-t003:** Hot stamping process parameters of the GH3625 sheet.

Parameter	Value	Parameter	Value
Transfer time of blank/s	8	Heating temperature of blank/°C	1000
Press stoke/mm	350	Blank-holder force/KN	120
Forming speed/mm·s^−1^	100	Tool temperature/°C	300
Cushion stroke/mm	100	Friction coefficient	0.45
Waiting time before ram motion/s	3	Dwell time/s	120

## Data Availability

Not applicable.

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
