# Peer review of "Study on Hot Tensile Deformation Behavior and Hot Stamping Process of GH3625 Superalloy Sheet"

_materials, 2023, doi:10.3390/ma16051927_

Round 1

Reviewer 1 Report

The main question was to study GH3625 sheet behavior at different temperatures and strain rates to design the most appropriate regime for hot stamping. The paper is of definite interest and provides the approach of raising the efficiency of hot stamping processes. The effect of holding time and temperature on grain growth was investigated, the model of grain growth was constructed and was applied for hot stamping of GH3625 alloy sheet. The reference are appropriate.

1. Please, check the spelling of coefficients’ numerical values (page 10, line 248).

2. Please, specify the dimensions of the billet prior to hot stamping process (section 3.5). Thickness was 2 mm? Correct me, if I am wrong.

3. Conclusions, page 16, line 353. When you mention “good prediction accuracy”, please, provide numerical values approving this statement.

Author Response

Dear Editors and Reviewers:

Thank you for your letter and for the reviewer’s comments concerning our manuscript entitled “Study on hot tensile deformation behavior and hot stamping process of GH3625 superalloy sheet”. Those comments are all valuable and very helpful for revising and improving our paper, as well as the important guiding significance to our researches. We have studied comments carefully and have made correction which we hope meet with approval. The review comments and the responses to each comment are listed in the word.

Reviewer 2 Report

Dear Editor,

In this study, hot deformation behavior of GH3625 superalloy sheet was investigated in order to understand its behavior in hot forming process. I think the article is a useful study in general. However, some revision processes are required. The article can be simplified by focusing on the effects of hot deformationin mechanical properties in general. Giving many equations has caused a complex appearance in some parts of the article.

Title can be updated. Only hot deformation data is available in the article. There is a little  study related to hot stamping.

In the introduction part, the literature study is insufficient. This section should be expanded. In addition, the innovative aspect of the material and the article should be emphasized.

GH3625 superalloy sheet material is Chinese standard or Chinese brand. It should be emphasized which material this material corresponds to in international norms. This will increase the readership and citation potential of the article.

There are areas that are difficult to see in Figure 1.

The gleeble device used in the tests should be mentioned in more detail.

Figure 5 should be given larger.

Figures should generally be of higher quality and larger.

What is the purpose of fitting 8th degree polynomial? (Figure 13)

References are outdated. There are many studies on hot forming and hot stamping in recent years.

Author Response

(The authors gave the same response as above.)

Reviewer 3 Report

Questions and comments requiring clarification

1.     I suggest not divide words. If it is possibile I recommend correct it in the whole of the manuscript. It will be more clear for readers in my opinion.

2.     Gleeble system is a metallurgical processses simulator. What do You mean  if You write isothermal Simulator? I suppose that the fact that test were carried out with the constant temperature value? Correct and write „metallurgical processses Simulator” please because the device enables also to provide the test with changing of temperature value during the test.

3.     Reffers to the text between Line no: 78-81. On what basis were selected heat treatment parameters described between line no 78-81? If from the technical literature, please provide a reference.

4.     According to the standard, flat samples for ten sile test must have the appropriate length of the base part depending on the thickness and width. Do the samples used comply with the relevant standards? Please provide a reference.

5.     Reffers to the figure no 5-7, 9-11and 13-16: If it is possibile use larger font on the diagrams axis descritpion and legend please.

Author Response

(The authors gave the same response as above.)

Round 2

Reviewer 2 Report

The revisions are enough for me. It can be accepted for publication if deemed appropriate by the editorial office.